# Computer-Aided Drug Design Boosts RAS Inhibitor Discovery

**DOI:** 10.3390/molecules27175710

**Published:** 2022-09-05

**Authors:** Ge Wang, Yuhao Bai, Jiarui Cui, Zirui Zong, Yuan Gao, Zhen Zheng

**Affiliations:** 1Medicinal Chemistry and Bioinformatics Center, Shanghai Jiao Tong University School of Medicine, Shanghai 200025, China; 2College of Stomatology, Shanghai Jiao Tong University, Shanghai 200120, China

**Keywords:** RAS inhibitor, computer-aided drug design, virtual screening, molecular docking, molecular dynamics simulation

## Abstract

The Rat Sarcoma (RAS) family (NRAS, HRAS, and KRAS) is endowed with GTPase activity to regulate various signaling pathways in ubiquitous animal cells. As proto-oncogenes, RAS mutations can maintain activation, leading to the growth and proliferation of abnormal cells and the development of a variety of human cancers. For the fight against tumors, the discovery of RAS-targeted drugs is of high significance. On the one hand, the structural properties of the RAS protein make it difficult to find inhibitors specifically targeted to it. On the other hand, targeting other molecules in the RAS signaling pathway often leads to severe tissue toxicities due to the lack of disease specificity. However, computer-aided drug design (CADD) can help solve the above problems. As an interdisciplinary approach that combines computational biology with medicinal chemistry, CADD has brought a variety of advances and numerous benefits to drug design, such as the rapid identification of new targets and discovery of new drugs. Based on an overview of RAS features and the history of inhibitor discovery, this review provides insight into the application of mainstream CADD methods to RAS drug design.

## 1. Introduction

RAS as GTPase is a binary switch that functions in cellular signal transduction. In normal situation, this function can be regulated precisely. However, mutations of RAS or its regulators keep RAS continuously active in RAS-related diseases, such as cancer and psychiatric disorders. Statistically, RAS mutations account for 25% of all human cancers. Although mutations of all three types of RAS members (NRAS, HRAS, and KRAS) can cause cancer, KRAS is the most common oncogene, accounting for 85% of all RAS mutations. KRAS mutations alone cause approximately one million deaths worldwide annually [1], specifically, 91% of pancreatic cancers, 42% of colorectal cancers, and 33% of lung cancers [2]. Hence, the optimal therapeutic strategy for RAS-related diseases is to discover new RAS inhibitors to effectively constrain the abnormal activation of RAS mutations. Although RAS has been called “undruggable” in recent decades because the surface of the RAS protein is smooth, the development of computer-aided drug design (CADD) has significantly facilitated the discovery of the specific RAS inhibitors.

CADD is important not only in leading compound discovery to predict the potential targets and compounds but also to evaluate biological competencies and optimize drug activity. Based on different structural data, CADD is generally divided into two strategies: structure-based CADD (SB-CADD) and ligand-based CADD (LB-CADD). SB-CADD prefers target proteins with high-resolution three-dimensional (3D) structures and the identification of binding sites [3,4]. Experimental determination or computational calculation provides these data for molecular docking and other SB-CADD methods, through which the interaction between the target protein and ligand molecules can be evaluated [5]. LB-CADD is an indirect drug design method based on a set of active ligands with certain structural characteristics. By modeling and similarity searching, potentially bioactive compounds can be discovered without knowing the 3D structure of the target protein. A quantitative structure–activity relationship (QSAR), a crucial LB-CADD algorithm, converts the chemical structure of the molecule into descriptors to perform statistical operations and quantitative analysis [6,7].

In terms of tactics, CADD can be divided into virtual high-throughput screening (vHTS) and de novo drug design [8]. The former requires an existing molecular database for screening, while the latter relies on generative models [9,10]. vHTS has been essential in industrial and academic drug discovery for decades [11]. Structure-based vHTS requires molecular docking to critically evaluate ligand–receptor affinity and simulate their binding patterns while screening specific biologically active compounds [4]. Ligand-based vHTS analyzes QSAR descriptors or other quantitative features of ligands while screening. The screening efficiency of vHTS depends on the precision of a certain method and could be enhanced by combining structure-based and ligand-based strategies [12]. De novo drug design, on the other hand, is a tactic that is more demanding and less popular than vHTS. It uses computational algorithms to generate compounds from scratch without a molecular database [10]. In structure-based de novo design, small fragments that match the binding site are created and then assembled into feasible compounds with a novel structure. Although rarely mentioned in RAS inhibitor discovery, de novo design may show its advantage in the future with broader exploration in chemical space and with the application of machine learning [13].

In general, both of these strategies or tactics have been used alone or in combination in drug discovery. This review provides insight into the applications of several common CADD methods in RAS drug design based on an overview of RAS features and the history of inhibitor discovery.

## 2. Biochemical Features of RAS

### 2.1. RAS in Normal Physiological Condition

RAS family proteins are encoded by three genes, namely HRAS, NRAS, and KRAS, which have highly homologous sequences and overlapping functions [14]. Among them, KRAS has splice variants representing two isoforms, KRAS4A and KRAS4B. Residues 1–166 form the G domain, which consists of six β-strands (β1–β6) surrounded by five α-helices (α1–α5) and ten connecting loops (L1–L10). Further structural studies at RAS have revealed P-loop (P or L1; residues 10–17), Switch I (residues 30–38), and Switch II (residues 59–76) on the surface of the G domain, which constitutes the active site for GTP/GDP and serves as interfaces for the binding of effector proteins and regulatory factors (Figure 1a) [15]. Most proteins in the superfamily of P-loop nucleoside triphosphate hydrolases (NTPases) contain a highly conserved sequence motif, P-loop. The P-loop keeps the GTP in an appropriate configuration for nucleophilic attack by a water molecule interacting with the Switch I and Switch II regions [16]. Twenty residues of RAS C-terminal (residues 167-188/189) form a hypervariable region (HVR).

RAS is a GTPase switch between the GTP-bound active state and the GDP-bound inactive state in several key signaling pathways that regulate cell growth, proliferation, and differentiation (Figure 1b) [17]. This switch is modulated by GTP-activated proteins (GAP) such as p120GAP, neurofibromin, type I neurofibromatosis (NF1) gene products, etc. [18] They facilitate GTP hydrolysis. There is a highly conserved “arginine finger” on GAP. Once this “arginine finger” moves into the active site of RAS-GTP, the intrinsic GTPase activity of RAS is greatly increased. RAS is also modulated by GMP exchange factors (GEF), such as the son of seven less homologue (SOS). They catalyze the loading of GTP to active RAS [19]. Activated RAS anchors to the cell membrane via the C-terminal CAAX box (C, cysteine; A, aliphatic amino acid; X, any amino acid) of the HVR region [20], which is post-translated by farnesyltransferase (FTase) or geranylgeranyltransferase (GGTase) for prenylation to dimerize or assemble “nanoclusters” of 5 to 10 monomers to regulate downstream signaling [21,22]. Following isoprene conversion at the CAAX cysteine, the RAS-converting enzyme (RCE1) performs proteolytic cleavage at the RAS AAX terminal, followed by carboxymethylation by isoprenylcysteine carboxyl methyltransferase (ICMT).

**Figure 1 molecules-27-05710-f001:**
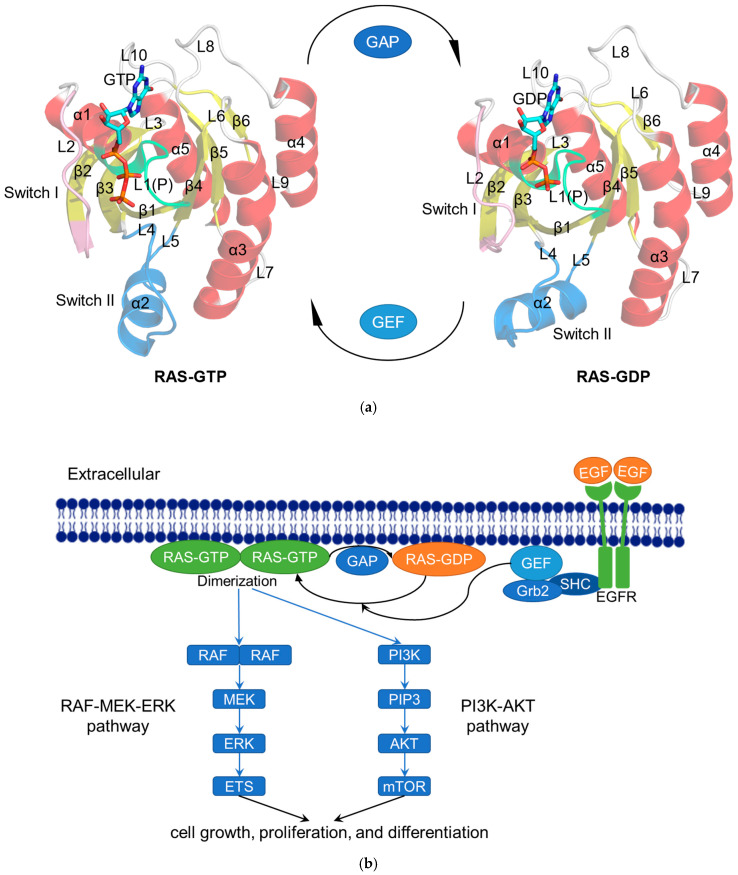
The RAS functions as a binary switch in normal state. (**a**) Cartoon representation of the crystal structure of RAS complexes: KRAS4B–GTP (modified from KRAS4B–GppNHp; PDB ID: 3GFT) and KRAS4B–GDP (PDB ID: 4LPK). The helices (α1–α5), strands (β1–β6), and loops (L1–L10) are shown in red, yellow, and gray, respectively. The P-loop (P or L1), Switch I, and Switch II regions are colored lime, pink, and blue, respectively. GTP/GDP are depicted by stick models [23]. (**b**) Schematic diagram showing the RAS-related signaling pathways. After activation by epidermal growth factor (EGF), the tyrosine kinase receptor EGFR recruits GEF such as SOS to the cell membrane via Src homology 2 domain containing (SHC) and growth-factor-receptor-bound protein 2 (Grb2) to activate RAS [24]. Subsequently, the activated RAS dimerizes and binds to the downstream RAF protein to regulate the MAPK signaling pathway (RAS–RAF–MEK–ERK pathway). The activated ERK is transported to the nucleus and then phosphorylates a number of transcription factors, such as erythroblastosis virus transcription factor (ETS), to ultimately regulate the cell cycle [25]. In another RAS–PI3K–AKT pathway, the activated RAS recruits PI3K to phosphorylate the substrate PIP2 and generate PIP3, whereupon PIP3 causes the sequential phosphorylation of AKT and mTOR to regulate cell proliferation [26].

### 2.2. RAS Mutations Trigger Signaling Dysfunction

KRAS G12 mutations are predominant (81%) in human cancers, followed by G13 (14%) and Q61 (2%) (Figure 2a) [1]. Glycine at codon 12 or 13 mutated to an amino acid other than proline results in steric hindrances that reduce the van der Waals forces between RAS and the GAPs and prevent the arginine finger of GAP from entering the GTPase site [27]. KRAS^G12C^ could prompt the exposure of the effector binding site and the nucleotide binding site, which can increase the intrinsic GTPase activity and thus upregulate signal transduction [28,29]. As an actuator of GTP hydrolysis, mutations of glutamine 61 disrupt GAP-mediated intrinsic GTP hydrolysis [27]. Other mutations such as A146V [30], T158A [31], and R164Q [28] cause the rapid dissociation of GDP (Figure 2b). Overall, most RAS mutations can disrupt the balance of the GTPase switch function and continuously activate downstream signaling pathways, leading to the uncontrolled proliferation of tumor cells.

Three main strategies have been proposed to treat RAS-related cancers: (I) targeting the signaling pathway upstream of RAS, such as vascular endothelial growth factor (VEGF), epidermal growth factor receptor (EGFR), or the phosphorylation of upstream regulatory kinases [32]; (II) targeting RAS downstream signaling, such as RAF [33], MEK [34], ERK [35], PI3K [36], AKT [37], or mTOR [38]; (III) directly targeting RAS itself. However, in the absence of a deep pocket for binding small compounds with high affinity, RAS has long been considered undruggable [39,40]. In recent decades, it has become apparent that advanced CADD technology can transform RAS into a promising druggable target. Inhibitors that directly target RAS can be divided into two categories: (i) targeting the plasma membrane localization fragment of RAS: farnesyltransferase (FTase) inhibitors disrupt the post-translational modifications of the CAAX tetrapeptide chain [41] and phosphodiesterase δ (PDEδ) inhibitors prevent PDEδ from binding farnesyl, thereby trapping RAS into the interior of cells [42]; (ii) targeting the activation domains of RAS: drugs binding to the α4–α5 region of RAS can disrupt RAS–effector interactions [43], and binding to the Switch II pocket (in GDP-bound inactive conformation) can block the interaction between RAS and SOS [44,45], whereas BI-2852 and BAY-293 have been shown to inhibit the interaction between KRAS and SOS1 with effective antitumor potency [46].

## 3. Application of CADD Methods in the Development of RAS Inhibitors

CADD has been increasingly important for the discovery of new inhibitors targeting RAS and its upstream or downstream signaling pathways. Based on high-resolution 3D apo or complex structures of RAS and its upstream and downstream proteins, SB-CADD is the optimal strategy for successful inhibitor discovery, especially vHTS in combination with molecular docking and molecular dynamics (MD) simulations. In addition, LB-CADD is also an essential strategy for inhibitor discovery that includes QSAR and pharmacophore modeling. More advanced computer algorithms, such as machine learning, are also promising for the discovery of RAS-related inhibitors.

### 3.1. Determination of the Target Protein Structure

To analyze the structural features of proteins, discover potential drug targets, screen potential drugs, and accelerate drug development, researchers need to determine the structures of target proteins. Many methods have been proposed for this purpose. The most traditional among them are nuclear magnetic resonance spectroscopy (NMR), X-ray crystallography, and cryo-electron microscopy (cryo-EM). Currently, there are approximately 400 open-access RAS structures in the Protein Data Bank (PDB) database [47], most of which were obtained using these methods. Although these approaches are widely used and have high measurement accuracy, they are time consuming and expensive. Therefore, the newly developed CADD method is crucial for predicting the structure of target proteins and can be used to discover potential drug binding sites.

Homology modeling is a common method for estimating the structure of target proteins and evaluating structural properties based on the homologous sequence of proteins [48]. Many applications provide homology modeling, such as Modeller and SwissModel [49]. RAS-association domain family (RASSF) 2 is a tumor suppressor protein interacting with KRAS, whose epigenetic inactivation through promoter hypermethylation is frequently detected in multiple mutant RAS-containing primary tumors. Since RASSF2 acts as a proapoptotic KRAS-specific effector, some RAS inhibitors take effect through its overexpression to promote apoptosis and cell cycle arrest [50]. The typical amino acid sequences were retrieved from the Uniprot database by Kanwal et al. [51]. Six templates were then observed by searching for templates based on the query sequence using the NCBI Basic Local Alignment Search Tool (PSI-BLAST). Finally, the 3D structure of the target protein was generated by comparative modeling using spatial constraint-based Modeller (9V15), I-Tasser, SwissModel, 3D-Jigsaw, and ModWeb, with quality checks on the ERRAT protein structure verification server [51]. Compared to traditional methods such as X-ray crystallography and NMR, homologous modeling has the advantage of being cost-effective and time-saving in predicting the 3D structure of proteins. However, the obtained 3D model is often improper and inapplicable when the homologous sequence is inadequate [48].

MD simulation is excellent for checking the quality of protein models by monitoring the atomic motion in real-time using Newton’s equation of motion. A reliability structure with the optimal statistical eigenvalues of molecular dynamics and thermodynamics is then determined [52]. For example, Prakash et al. used two sets of classical MD simulation to determine the stability of the predictive model to be optimized [53]. However, the MD simulation requires high computational power to predict the protein structure in a given force field with multiple intramolecular interactions.

The template-based protein–protein interaction complex structure prediction algorithm, PRISM, is useful for predicting the assembly of polymers or complex protein structures. PRISM compares the known protein–protein interface structure data with the protein interactions to be measured to derive the shape of the protein interaction region of interest [54], which drastically reduces the computational effort and research cost compared to ab initio methods. To study GTP-dependent KRAS dimerization, Muratcioglu et al. [55] discovered two primary dimer interfaces dependent on the binding properties of KRAS to GTP by performing a series of conventional experiments including dynamic light scattering (DLS), isothermal calorimetry (ITC), microscale thermophoresis (MST), fluorescence spectroscopy, Forster resonance energy transfer (FRET), and NMR. From then on, PRISM was used to predict the dimer structure.

In recent years, the burgeoning field of artificial intelligence (AI) has shown that it is possible to accurately predict protein structure with the increasing abundance of protein databases. Machine learning and deep learning algorithms have been used to predict protein structures, which are highly efficient and accurate compared to traditional CADD methods [56]. Li et al. [57] employed the well-known AlphaFold [58] to obtain models of 145 members of the RAS superfamily and discover the accessible cysteines in the allosteric pockets. It is expected that AI will be widely used for protein structure prediction in the near future and will contribute greatly to the development of molecular biology and pharmacy.

### 3.2. Identification of Binding Sites

In drug discovery, the identification of potential binding sites for small molecules is critical for the 3D structural model of the target protein. This process can be achieved by using a variety of methods to calculate and identify binding sites. Evaluating the energy, volume, and shape of potential binding sites can reflect the binding capacity of drugs [51,59].

New algorithms such as Phosfinder [60], LPIcom [61], Sitehound-Web [62], and GenProBiS [63] provide the recognition function of ligand binding sites in protein structures. In the study on RASSF2, Kanwal et al. [51] derived the potential binding sites of RASSF2 via Sitehound-Web after predicting the 3D structure of the RASSF2 protein. The efficiency of the binding sites can be verified by measuring the energy range and volume of the pockets. These web servers can be used to conveniently discover the potential binding sites of the predicted structures.

Probe-based molecular dynamics (PMD) simulation is a common method for discovering potential binding sites of target proteins by adding probe molecules to the conventional MD simulation process. Potential drug targets are identified based on the frequency of contact between the target site and the molecular probe relative to the druggability of the site [64]. In a study on the binding hotspots of KRAS, Prakash et al. [65] investigated the surface of KRAS using PMD simulation to evaluate the probability of interaction with organic molecules. Among eight potential druggable sites, five constitute the ligand binding pockets parallel to experimental results (Figure 3, Table 1). Overall, PMD simulation can predict the potential binding sites of various proteins by simply modifying the probe.

In contrast to PMD simulation, the fragment-based approach, FTMAP, uses relatively fixed probes. It correlates the druggability of the pocket with its propensity to bind these probes by rigidly docking each probe to generate thousands of binding sites and obtaining the final conformations through clustering and minimal free energy [66]. In identifying new allosteric sites on RAS, Grant et al. [67] used FTMAP to discover three new potential binding sites: P1, P2, and P3 (Figure 4, Table 2).

The multiple solvent crystal structures (MSCS) method employs organic solvent molecules to detect and characterize ligand binding sites on proteins. In MSCS, the X-ray crystal structure of the target protein is dissolved in different organic solvents. Organic solvent molecules that accumulate at specific sites on the protein indicate that they are potential sites for molecular interactions [68]. In a hot spot analysis of RAS GTPase surface binding sites, MSCS helped Buhrman et al. [69] to identify the potential binding sites of HRAS-GppNHp protein–protein interactions. Eight potential binding sites were obtained from different conformations under the crystal structure in the solvent (Figure 5, Table 3). However, MSCS is limited by its reliance on the X-ray crystal structure of the target protein.

In addition to predicting the structure of the target protein, AI can also be applied to accurately discover the potential binding sites on the target protein [70]. Discovering potential binding sites with AI will predict the binding ability of the binding sites to the ligands by narrowing the selection range of the target ligands, which is a primary direction for drug design. Table 4 shows the application of CADD in RAS-related structure and binding site identification.

### 3.3. Virtual Screening

Virtual screening (VS) automatically searches a small-molecule library for structures that potentially bind well to the target biomolecule [71]. VS is a combination of several techniques based on high-throughput screening from millions of lead compounds, i.e., vHTS, to provide a solid foundation for further work. Although the continuous improvement of computer hardware has led to an unprecedented increase in computational power to screen as many compounds as theoretically possible, it is more recommended to construct optimal small-compound libraries and improve the computational speed and hit rate with various optimization algorithms for virtual drug screening strategies.

VS can be broadly divided into two types of screening strategies: receptor (structure)-based and ligand-based approaches [71]. Here, receptor and ligand stand for target proteins (drug targets) and small molecules (drugs), respectively. Structure-based virtual screening (SB-VS), also known as receptor-based virtual screening (RB-VS), is progressed inseparably from known protein structures and binding sites. In particular, this strategy involves molecular docking, structure-based pharmacophore modeling, molecular dynamics simulation, etc. SB-VS sometimes has to enable rapid high-throughput screening at the expense of scoring accuracy, so the induced fit effect and solvation effect are generally ignored. Ligand-based virtual screening (LB-VS) predicts the potential structure of candidate drugs from a set of active compounds known to bind to specific sites on the target protein. The features of screened ligands are extracted, such as structural conformation, charge distribution, and physicochemical properties. Then, the feature library, including molecular fingerprints, pharmacophore models, or matching ideal compounds, is constructed by QSAR. This strategy is mainly used when the structure of the target protein is unknown or when the target site is predicted in reverse, while a minor structural change can lead to a drastic change in molecular activity.

For RAS inhibitor discovery, an emerging “hybrid virtual screening” method has been employed to extract information from the global features of existing ligand–receptor complexes for similarity-matching to reduce the false positive rate and increase computational efficiency [72,73]. KRAS^G12D^ mutations resulted in a conformational exchange of exposing the Switch I region to continuously activate the GTP-bound form, which was more susceptible to bind to GEFs (e.g., SOS1). GTPase activity subsequently decreased, leading to the aberrant activation of downstream signaling [74,75]. Hashemi et al. developed a strategy to inhibit the guanylate cycle by competitively preventing the inactivated state of KRAS^G12D^ (iKRAS^G12D^) from binding to SOS1 [76]. In the first round, flavonoids were selected as ligand sets for LB-VS. In the second round, on LigandScout software, hybrid virtual screening strategy was used to build a pharmacophore model by extracting basic interaction parameters involving hydrogen bonds, salt bridges, aromatic rings, and hydrophobic regions from the existing co-crystal complexes of active inhibitors binding to the SOS1–KRAS interface. Finally, nine hit compounds were screened out of 250,000 compounds in the National Cancer Institution (NCI) database for subsequent molecular docking and molecular dynamics simulation screening. In search of more effective inhibitors, the nine candidates were then used to generate new ligand sets to repeat the process in the PubChem database. Auriculasin (−9.8 kcal/mol) was finally identified with a higher affinity than the reported inhibitor DCAI (−5.2 kcal/mol) for inhibiting the SOS1–KRAS^G12D^ interaction [77].

Due to RAS’s undruggability, the post-translational modification (PTM) of RAS is considered a drug target, such as farnesyltransferase. As a potential inhibitor, theaflavin (Vina score = −12.2 kcal/mol) is identified as an alliance of virtual screening with Autodock Vina based on the Lamarckian genetic algorithm, molecular docking, and molecular dynamics [78]. After PTM, RAS localization and transport regulated by the prenyl-binding protein PDEδ is also promising for drug intervention [79]. NHTD targeting the hydrophobic prenyl pocket of PDEδ (Glide XP score = −12.77 kcal/mol) was discovered by Leung et al. [80], who applied an extra precision (XP) docking scoring to 1.3 million compounds with Glide [81]. Aiming at the interface of the KRAS4B–PDE6δ crystal complex as a binding pocket, VS in collaboration with MOE Dock was also performed to search for compounds that prevent the mutant KRAS from being released from the stable complex and are candidates for the treatment of pancreatic ductal adenocarcinoma [82]. MEK1 inhibitors targeting the RAS downstream pathway were identified by a cooperation of GOLD and CDOCKER and finally adopted by Glide XP in collaboration with HypoGen pharmacophore modeling as a scoring method with a low root mean square deviation (RMSD) to improve screening reliability and efficiency. After screening the SPECS library of more than 200,000 compounds, the candidate MEK1 inhibitor was validated by biochemical and cellular assays (IC_50_ = 3.5 μM) [83]. Multiple approaches, including Fischer’s randomization, test set prediction, and decoy set confirmation, have been used in many studies to assess the quality of pharmacophore modeling and analyze costs. Glide XP scoring was also used to identify catechin (Glide XP score = 10.98 kcal/mol) as another potential MEK1 inhibitor for the treatment of melanoma [84]. In this study, the known MEK1 inhibitory ligands in Ligand.Info were screened for analogue compounds to build a drug library for molecular docking in SB-VS rather than LB-VS. Due to flexibility in ligand selection, low computational cost, and methodological versatility, LB-VS has been extensively used despite the known target protein structure in most cases. VS is considered to be advantageously compatible with multiple CADD screening strategies in the search for RAS inhibitors.

### 3.4. Molecular Docking Studies

As the most common method in the structure-based drug design of CADD, molecule docking is widely used in RAS-targeted drug discovery. The basic principle of molecule docking is to give a prediction of the ligand–receptor complex structure by computation methods. To achieve this aim, there are two key steps: sampling and scoring. The former targets ligands and active sites of proteins; moreover, it estimates experimental binding modes; the latter evaluates binding affinity through a scoring function (with various assumptions and simplifications) [85].

The earliest reported docking method is called “rigid body docking” derived from Fischer’s “lock-and-key assumption”. The essence of rigid body docking is that both ligands and receptors are treated as rigid bodies and that the affinity between ligand and receptor is positively related to the geometric fit between their structures [86]. This method is still used for macromolecular interactions due to its simplicity and feasibility. However, in practical tests, rigid body docking procedures such as ClusPro produced obvious disadvantages, such as fewer hits as the top 1 prediction and the lower accuracy of the generated models [87]. Over time, the theory of “induced-fit” was developed. It states that the active site of the enzyme is non-rigid. The substrate can induce a corresponding conformational change of the active site of the enzyme, and the relevant groups rearrange the correct orientation so that the enzyme and substrate fit together to form an intermediate complex, causing a reaction [88]. As a developmental method, “flexible docking” allows the conformational changes of small molecules or targets to precisely examine intermolecular recognition by matching spatial shape and energy. The binding capacity is ultimately determined by the change in binding free energy (ΔGbind) during the formation of the complex with indispensable kinetic considerations [89]. To simplify biological macromolecular dynamics, the assumptions of additivity and transferability have been employed in force fields instead of electronic degrees of freedom. Most classical force fields focus on five terms: the bond deformation and the angular geometry (stretching/compression of bonds, angular bending), rotation around some dihedral angle (torsion), the so-called nonbonding, the electrostatic interactions, and the dispersive interactions, as well as the repulsion because of atoms overlapping (van der Waals forces). The extended complex force fields include atomic polarizability and coupling forces, for instance, cross-coupling between bonds and angles [90].

In the history of RAS-targeted drug discovery, molecular docking technology has screened an indirect inhibitor to inhibit RAS nucleotide exchange. By constructing a series of KRAS conformers containing infrequencies and performing blind docking with AutoDock4.2, a bicyclic diterpene lactone from andrographis paniculate andrographolides (AGP) was identified. Its benzylidene derivatives can bind to a transient pocket on KRAS to block the exchange of GDP-GTP and thus inhibit the signal transduction of KRAS. The successful inhibition study not only suggests that nucleotide exchange factors are required for RAS signal transduction but also demonstrates that the inhibition of nucleotide exchange is a practical approach to abolish the function of oncogenic RAS mutants [91].

In RAS post-translational modification, processing required three enzymes, as mentioned earlier, so targeting one of these enzymes is undoubtedly a promising strategy for inhibiting the process [92]. Molecular docking uses the crystal structure of FTase as a template to clarify the interaction between antroquinonol and the CAAX box of FTase after verifying the inhibition of isoprene transferase by antroquinonol in vitro. In conclusion, antroquinonol can inhibit the activity of prenyltransferase to restrict the activation of RAS and RAS-related GTP-binding protein, leading to the activation of autophagy and death of the cancer cell [93].

As a molecule in the RAS–RAF–MEK–ERK signaling pathway, BRAF protein kinase mutating in approximately 7% of human cancers has been manifested as elevated kinase activity and considered an important therapeutic target for inhibition. A study identifying 18 compounds targeting BRAF through virtual screening and ELISA confirmed that compound 1 efficiently inhibits BRAF kinase. Moreover, molecular docking clarifies the docking conformation of compound 1 in the active site of BRAF and deduces the scaffold based on the key of hexahydropteridine moiety. This conjecture has been confirmed by ELISA with homologous compound **19**. After a series of in vitro experiments with analogs of compound **19** (**19**–**33**), compound **24** exhibits the most potent inhibitory activity [94].

Since RASSF2 as a potential tumor suppressor gene promotes apoptosis and cell cycle arrest, Modeller (9V15) and online servers (I-Tasser, SwissModel, 3D-JigSaw, ModWeb) generated the 3D structure of RASSF2 based on homology modeling to identify its top 10 binding pockets ranked by energy. Furthermore, AutoDock Vina and AutoDock4 recognize the ligands of RASSF2 that regulate the normal activity of RASSF2. Finally, as stabilized RASSF2 compounds, ANP (phosphoaminophosphonic acid adenylate ester) and GNP (phosphoaminophosphonic acid guanylate ester) can serve as lead compounds for further studies targeting the RASSF family [51].

Initially, molecular docking is used to study the interaction mode between molecules, which plays an irreplaceable role in clarifying the mechanism of intermolecular interaction, discovering essential parts and guiding the synthesis of lead compounds. The rapid development of molecular docking enables the automatic screening of numerous compounds, which has been one of the mainstream methods of CADD. In addition, molecular docking also plays a vital role in drug repositioning and adverse reaction prediction. In summary, molecular docking is rapidly developing as a promising CADD method.

### 3.5. Molecular Dynamics (MD) Simulation

The MD simulation is endowed with a temporal dimension in which the dynamic interactions at the atomic and molecular levels can be traced over a period of time [95]. Compared to the static function of the structure determined by X-ray crystallography or cryo-electron microscopy, the MD simulation builds the time function of position and velocity for each atom with ideal environmental variables and initial flexible conformations evolving to final molecular conformations with lower energy or ligand–acceptor interaction patterns. In the MD simulation, a number of conditions should be tightly controlled, including the incorporation of mutations/modifications, the selection of ligand receptors, the imposition of perturbations, etc. Since the initial conditions are limited, the errors caused by the integration process in the simulation accumulate over the time dimension and cannot be completely eliminated. Therefore, it is necessary to improve the accuracy by optimizing the algorithms, exerting appropriate molecular mechanical force fields, and setting suitable parameters. In this regard, the three basic parameters determine the accuracy, such as the simulation range (regional treatments are often required), the time step (should be smaller than the minimum period of oscillation of the particle system, about 10^−15^ s), and the total time duration (must cover the period of natural interaction dynamics, 10^−9^–10^−6^ s or longer).

In classical MD simulation, the trajectories of molecules are traced by solving the Newtonian equations of motion of the interacting particle system to approximate the quantum mechanical model: the kinematic parameters of the atom are determined via interaction forces with given atomic coordinates and random initial velocities. The crucial kinetic steps are often located in transition states with high free energy and are difficult to be sampled, especially in complicated protein systems with heterogeneous states, complicated interactions, and undefinable solvent effects. To solve these problems, various algorithms should be carefully selected and optimized. The nudged elastic band (NEB) algorithm can calculate the transition state of protein structures within the minimum energy path (MEP) between different conformations [96]. Similarly, Pande et al. [96] used the well established Markov state model (MSM) method to describe long-time dynamics based on the transformation between Markovian substable conformations [97]. The accelerated MD (aMD) simulation method has been used in previous studies to calculate the short time scale in classical MD simulation [98]. Various MD simulation programs, such as GROMACS, AMBER, CHARMM, and NAMD, can evaluate the ligand–receptor binding properties in terms of RMSD, root-mean-square fluctuation (RMSF), interaction forces, and the energy of the complex system, etc. The force field of MD represents a potential energy function consisting of the functional form and the parameter sets. The parameter sets depend on the atom types of the MD molecular systems and are transferable based on the structural similarity of the molecules. For apo structures and polymers of macromolecular systems in biochemistry, optimized potentials for liquid simulations (OPLS), assisted model building with energy refinement (AMBER), and chemistry at Harvard macro-molecular mechanics (CHARMM) are common all-atom force fields with higher simulation accuracy, while Groningen molecular simulation (GROMOS) is a united-atom force field with higher computational efficiency.

MD simulation can break the bottleneck in determining protein structures and molecular interactions, especially in drug discovery [99]. In the SOS-induced nucleotide exchange process of the RAS system, MD simulation identifies the stable binding site of SRJ23 in the KRAS4B–SRJ23 (benzylidene derivative of AGP) complex [100,101]. Based on experimental evidence that the Src-induced dual phosphorylation of KRAS Y32/64 disrupts the GTPase cycle to interfere with RAS downstream signaling [102], MD simulation has revealed the complex process of unphosphorylated or phosphorylated KRAS4B in GAP, SOS, and RAF in the GTP/GDP-bound states. The dual phosphorylation of KRAS4B alters the nucleotide binding site conformations and generates perturbations at the catalytic site, resulting in the expansion of the GDP binding pocket and the latency of the intrinsic hydrolysis of RAS GTP. This has identified RAS phosphorylation as a drug target [103]. As a covalent inhibitor targeting the Switch II allosteric pocket (SII-P) for KRAS^G12C^, AMG-510 (SotoRASib) is the first FDA-approved drug discovered by MD simulation. Using all-atom simulation on a long time scale (75 μs), in MD simulation, Pantsar and colleagues [104] found that AMG-510 remains stably bound to SII-P during switch swing, rather than fixing KRAS switches based on crystal structure. With the MD simulation, AMG-510 also explains the interaction with KRAS of PTM [104]. The interaction mechanism and kinetic parameters between KRAS^G12C^ and another covalent inhibitor, ARS-853, were also revealed with molecular docking and MD simulation [105]. There is a challenge that the dimerization or oligomerization of RAS with PTM on membranes can only be resolved with recombinant lipid membranes or nanodiscs [55,106,107]. Therefore, MD simulation plays a pioneering role in discovering new potential drug targets and strategies by revealing the interface interaction and energetic information of RAS dimer or other RAS-related pathway proteins at the atomic level [53].

MD simulation is also a robust tool for discovering allosteric binding sites. Allostery can modulate protein structure and activity by binding an effector to an allosteric site instead of the orthosteric site [108]. Therefore, the discovery of allosteric sites [109,110,111,112], the exploration of the allosteric mechanism [113,114,115,116], and the targeting of allosteric sites for drug discovery [117,118,119] are of great importance. In combination with the transition pathway generation algorithm and MSM analysis, MD simulation helps to identify several key conformational substates in RAS deactivation hydrolysis and a novel potential allosteric binding site for inhibitors to block downstream signaling effects [99]. NS1 is a peptidomimetic that binds to the variable configuration site of RAS to inhibit RAS dimerization and prevent the abnormal activation of the downstream RAF–MEK–ERK pathway. However, the affinity of NS1 for HRAS is reduced by the HRAS^R135K^ mutation. In a 200-ns MD simulation with dynamic network analysis and investigation of the overall architecture of the allosteric network of HRAS, Ni et al. [120] found that HRAS^R135K^ disrupts most of the key interactions at the interface of the wild-type HRAS–NS1 complex and abrogates the original allosteric regulation. There are studies on the allosteric effects of KRAS as the regulated or the regulator. Using aMD and allosteric pathway analysis, the mechanism of the allosteric activation of PI3Kα by KRAS4B was elucidated to the extent that KRAS4B binding disrupts the interaction between the p110 catalytic subunit and the p85 regulatory subunit of PI3Kα. This disruption leads to the exposure of the kinase domain of PI3Kα, which facilitates its membrane binding and substrate phosphorylation [114]. Li et al. propose the mechanism of long-range allosteric regulation from Sprouty-related, EVH1 domain-containing protein 1 (SPRED1) to KRAS in the SPRED1–GAP (NF1)–RAS ternary complex via aMD and MSM analysis. NF1, acting as a scaffold, transfers the allosteric effect from the SPRED1 side to another (KRAS binding site), resulting in the restricted conformational change of the NF1 catalytic center for RAS hydrolysis but steadier KRAS–NF1 binding. Overall, SPRED1 enhances RAS–GTP hydrolysis [121]. The long-range allosteric regulation also exists in the catalytic KRas4B^G13D^ (nucleotide-free)–SOS–KRas4B^G13D^–GTP ternary complex. According to the MD simulation result, the binding of KRas4B^G13D^–GTP at the distal allosteric site of SOS increases the affinity between the catalytic KRas4B^G13D^ and SOS and promotes the removal of Switch I from the nucleotide binding site. These facts lead to a higher nucleotide exchange rate and generate more GTP-binding RAS for allosteric regulation as a positive feedback loop [122].

In general, MD simulation can firstly reveal the mechanism of the existing drugs at the atomic level, create new models for the potential drug–protein or protein–protein interactions, and design the precursor compounds. Second, the MD simulation can predict ligands as agonists or antagonists, with spatial resolution at the atomic level and temporal resolution at the femtosecond level. Third, the MD simulation can explore the allosteric sites and capture the temporal phase of the effect at the kinetic level and the receptor binding affinity. Finally, MD simulation can provide a more realistic and detailed atlas of pharmacodynamics and is more useful for drug design. Consequently, MD simulation is very helpful for the discovery of promising RAS inhibitors.

### 3.6. Quantitative Structure–Activity Relationship Study (QSAR)

QSAR is created by combining mathematical methods of empirical equations commonly used in physical chemistry based on the traditional structure–activity relationship between molecular structure and properties, such as molecules with similar structures having similar properties. Briefly, QSAR consists of the following core steps: experimentally determining the data for various compounds to construct training and test sets; computationally converting the structural formulas into the descriptor data for statistical operations, a process known as the acquisition of molecular descriptors; establishing a statistical model between the molecular properties of interest and the molecular descriptors of the training set; evaluating the obtained model according to various indicators; and attempting to explain the model from a mechanism. Various molecular descriptors reflecting different levels of chemical structure representation have been proposed as the core of QSAR. These levels range from molecular formulas (so-called 1D) and two-dimensional structural formulas (2D) to three-dimensional conformational formulas (3D) and higher formulas that take mutual orientation and time dependence into account (4D or higher) [6].

The 2D descriptor mainly defines the connectivity of atoms in molecules according to the existence and nature of chemical bonds, which is also called topological representation. This representation contains valuable and straightforward information about the molecular structure and has the advantage of being invariant to the rotation and translation of molecules. Although 2D descriptors cannot be used as unique representations without reconstructing molecules, they can characterize molecules with higher discrimination with well-defined ordered sequences [123]. Furthermore, 3D descriptors based on biological selectivity result from highly specific interactions between the target and ligands, such as hydrogen bonds. The ligand preferences arise mainly from non-covalent field effects imposed by the spatial proximity. The systematic sampling of field differences, such as the CoMFA formulation with the classical and dominant comparative molecular field, provides molecular descriptors suitable for QSAR. The main challenge in performing CoMFA is the alignment protocol for selecting conformation and orientation of the ligands in the training and test sets, which is cumbersome and expensive. However, new QSARs, such as topologically heterogeneous protocols, dramatically simplify reliable predictability [124]. Overall, 3D-QSAR research requires the structural alignment of compounds as the most critical step. Being related to the alignment protocol, the major obstacle in performing CoMFA is precisely the selection of ligands for the training and test groups, as well as the selection of the conformation and orientation of each ligand. In practice, this task often becomes slow and tedious and somewhat temporarily requires higher statistical standards (such as q2) [123]. To overcome this bottleneck, QSAR usually works with with other techniques such as molecular docking.

As a member of the RAS activator family (RAF), BRAF mutations exhibit markedly increased kinase activity and a high degree of disease severity. The BRAF V600E mutation, one of the most known oncogenic protein kinase mutations, represents an excellent potential therapeutic target. Due to the collaboration between molecular docking techniques and reliable CoMFA and CoMSIA models, three different V600E BRAF inhibitor datasets were generated based on a dataset of 125 compounds using receptor-guided alignment methods and database alignment. Both models show good statistical values and are validated by y-randomization tests. Finally, the newly predicted structure (IIIa) shows a higher inhibitory potency than the previous active compounds in the series (pIC_50_ = 6.826) [125].

### 3.7. Pharmacophore Modelling

The pharmacophore is a concept that represents the structural characteristics indispensable for ligand–target interactions [126]. According to the official IUPAC definition: a pharmacophore is the ensemble of steric and electronic features that is necessary to ensure the optimal supra-molecular interactions with a specific biological target structure and to trigger (or to block) its biological response [127]. The pharmacophore includes a range of hydrogen bond acceptors and donors, acidity, alkalinity, nucleophilicity, or electrophilicity of the functional group [128]. Based on the pharmacophore modeling of active ligands, vHTS can screen compounds with similar pharmacophore properties. If a compound has multiple pharmacophore features described in the pharmacophore modeling of active ligands, it is a multitarget compound [129].

Pharmacophore modeling in conjunction with vHTS is widely used as a reliable and rapid ligand-based CADD approach to discover inhibitors of RAS upstream and downstream molecules [128]. RAF kinase inhibitor protein (RKIP) is a critical regulator of the RAS–RAF–MEK–ERK signaling pathway. In recent research on RKIP inhibitors by Parate et al., a pharmacophore model with a series of hydrogen bonds, hydrophobic groups, and aryl rings of locoastatin (the most potent RKIP inhibitor known to date) was created [130]. Based on the pharmacophore model, compounds that have an analogous structure can be selected from the library by vHTS. As a result, the model has assigned a total of 2557 compounds out of 14,492 compounds in the Marine Natural Product Library. By optimizing the model, the number of compounds was significantly reduced, to 134 for further research.

Pharmacophore modeling was also applied to the structural orientation of QSAR modeling. A reliable 3D QSAR model was established by pharmacophore models with similar structural properties and molecular comparison for RKIP inhibitor discovery by Xie et al. [131]. Moreover, pharmacophore modeling also helps in the discovery of inhibitor targets RAS upstream and downstream molecules such as PI3K-α and PKC-η [59] (Table 5).

### 3.8. Other CADD Applications

Although vHTS is widely used, the de novo design of RAS inhibitors still alternatively shows a promising future ahead. Recently, proteins such as fluorescence-activating β-barrel were developed by de novo drug design [137]. A functional RAS-binding domain with extreme thermostability was identified by another de novo sequence redesign model called ABACUS by Liu et al. [138]. The de novo sequence redesign model does not suffer from a restrictively cumulative effect for future directions.

Currently, mainly qualitative or semi-quantitative methods are used in MD simulation to calculate the binding affinity with accurate free energy. Accurate free energy prediction methods, such as alchemical free energy method (AFEM) and absolute binding free energy (ABFE), greatly improve the efficiency of CADD, although it is extremely demanding and costly [139]. Free energy prediction models that are more accurate and efficient than molecular docking and less computationally demanding than AFEM, including Poisson–Boltzmann surface area (MM /PBSA) and Born surface area (MM /GBSA) generalized molecular mechanics, have already been used in current research, such as the discovery of RAS inhibitors [140]. The discovery of allosteric binders of RAS can also be empowered with free energy prediction methods. For example, the naphthyridinone scaffold was identified as a novel covalent allosteric binder for KRAS^G12C^ in free energy perturbation models [141].

As a more promising CADD method, machine learning (ML) algorithms, such as neural networks and the transformer, are developing suitable models to predict target protein structure and discover potential compounds. Other CADD methods, including molecular docking, QSAR, and pharmacophore modeling, also benefit from machine learning algorithms [142,143]. In molecular docking, ML is used for scoring functions that translate protein–ligand interactions into descriptors. In this way, effective scoring function models such as a NN Score and a RF Score can be built. In QSAR and pharmacophore modeling, ML improves the accuracy of molecular comparison and descriptor identification. It is foreseeable that machine learning algorithms with high efficiency will be increasingly used for RAS inhibitor discovery in the future.

## 4. Conclusions

RAS was once referred to as an “undruggable target” because of the special structural characteristics of RAS, the complexity of signaling pathways, and the drug resistance of RAS mutant tumors. RAS proteins have high intrinsic affinity for their GDP and GTP substrates and lack distinct pockets in their catalytic domain for binding compounds. However, thanks to the relentless efforts of researchers and advances in CADD, inhibitors have consistently shown satisfactory effects in experiments. To date, only a few small molecules have been able to covalently bind to KRAS^G12C^. Since the HVR mutations of RAS cause cancer, they can be used to develop small-molecule inhibitors targeting specific oncogenic RAS mutants. Recently, phosphorylation at multiple residues was identified to regulate the activation of RAS, which may be a new target for therapeutics against RAS diseases.

In recent years, FDA-approved KRAS^G12C^ mutation inhibitors (AMG 510, Sotorasib) have been introduced. The existing pan-RAS inhibitors, such as compound 3144, have unavoidable toxicity and off-target activity [144]. However, the discovery of RAS inhibitors is still a challenge. First, the discovery of specific inhibitors targeting other alleles is needed to provide a personalized medical approach, such as the common KRAS variants KRAS^G12D^ and KRAS^G12V^. Second, the effect of allele-specific inhibitors as monotherapy may be limited. Therefore, it is necessary to combine them with other inhibitors because cancers are often raised by multiple mutations [145]. Therefore, advanced knowledge of the pharmaceutical effects of other inhibitors on RAS alleles is necessary to develop strategic combination therapies. Combination therapy with solid therapeutic effects and relatively low toxicity is required. Third, further exploration of the mechanism of drug resistance by gene mutation and histological transformation is needed [146]. The rapid development of graphical computing, cloud computing, artificial intelligence technologies, and improvements in software and algorithms make this possible. Then, VS has become more diverse and high-throughput. MD simulation has even been extended to the millisecond level, accompanied by higher precision and lower consumption. QSAR and pharmacophore models have gained accuracy and efficiency as ML algorithms have improved. Overall, with increasing accuracy, CADD is becoming an indispensable component in RAS inhibitor discovery and other areas of drug discovery.

## Figures and Tables

**Figure 2 molecules-27-05710-f002:**
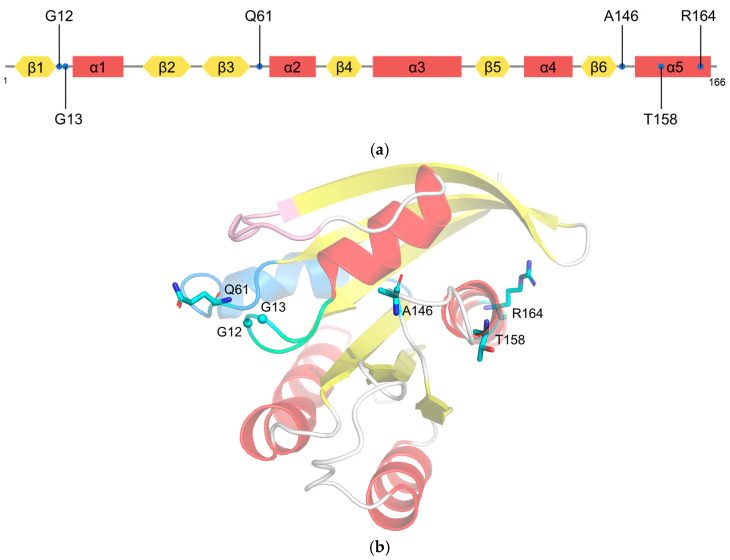
Mutations on KRAS. (**a**) Schematic diagram showing the positions of KRAS mutations; (**b**) stick representation showing six residue mutations mapped on the cartoon representation of the crystal structure of KRAS.

**Figure 3 molecules-27-05710-f003:**
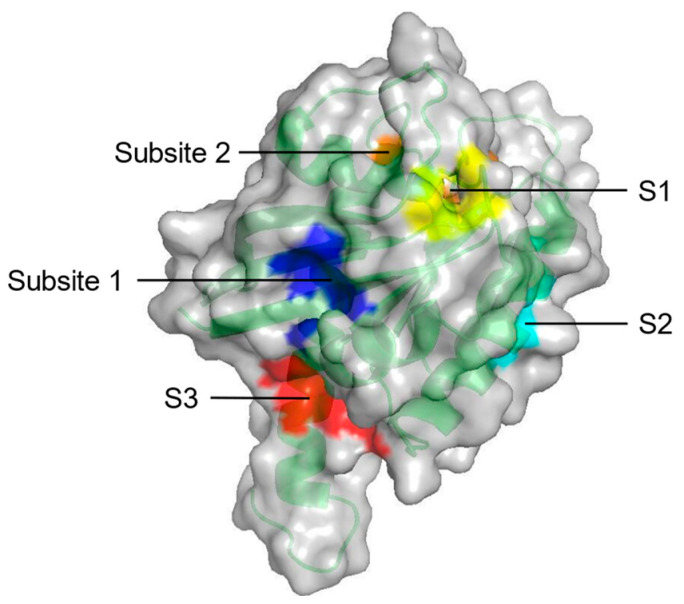
Surface representation of five potential druggable sites (S1–S3, Subsite 1 and Subsite 2) on KRAS from PMD simulation (PDB ID: 4DSO).

**Figure 4 molecules-27-05710-f004:**
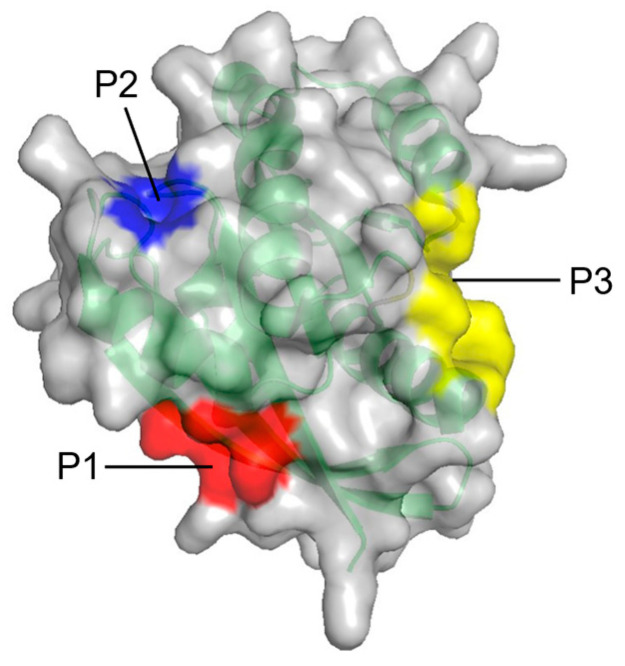
Surface representation of three potential allosteric sites (P1–P3) on RAS from FTMAP.

**Figure 5 molecules-27-05710-f005:**
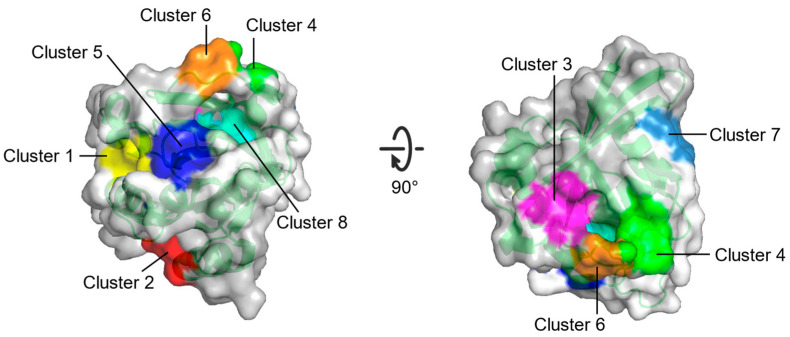
Surface representation of eight potential binding sites (Cluster 1–Cluster 8) on HRAS from the MSCS method.

**Table 1 molecules-27-05710-t001:** Constituents and location of experimentally identified pockets composed by potentially druggable sites (S1–S3, Subsite 1 and Subsite 2) from PMD simulation on KRAS.

Pocket	Constituents	Location
S1 + Subsite 1	V7, L56, M67, K5, D54, T74, Y71, E37, D38	In the core β-strand region behind Switch II
S2	V7, V9, G60, F78, M72, Q99, I100	Near Switch II and α3
S3	D105, S106, D107, D108, M111, E162, Q165, H166	Between L7 and α5
Subsite 2	D30, D33, D38, S39, Y40, I21, I36	At the back of Switch I

**Table 2 molecules-27-05710-t002:** Consist and location of three potential allosteric sites (P1–P3) on RAS from FTMAP.

Site	Constituents	Location
P1	K5, L6, V7, S39, D54, I55, L56, M67, Q70, Y71, M72, R73, T74, G75	Between β1–3 and α2
P2	Q61, E62, E63, Y64, S65, F90, E91, D92, I93, H94, H95, Y96, R97, E98, Q99	Between L2, α2, and α3
P3	R97, K101, E107, D108, V109, P110, M111, S136, Y137, G138, I139, P140, R161, E162, I163, R164, K165, H166	Between L7, L9, and α5

**Table 3 molecules-27-05710-t003:** Information about the eight potential binding sites (Cluster 1–Cluster 8) on RAS from FTMAP.

Site	Consist	Location
Cluster 1	R68, Q95, Y96, Q99, D92, E62, R68, D92, Q95, Y96, Q99, R102	Between switch II and α3
Cluster 2	H94, L133, S136, Y137, F90, E91, I93, H94, L133, Y137	Between α3 and α4
Cluster 3	S17, I21, Q25, H27, V29, D33, T35, D38, Y40	Opposite to Switch I relative to gppnhp
Cluster 4	F28, D30, K147	Near L8
Cluster 5	A11, G12, N86, K88, S89, D92	Between P-loop and N-terminus of α3
Cluster 6	D30, E31, Tyr32, GppNHp	Near N-terminus of switch I
Cluster 7	L23, N26, K42, V44, V45, R149, E153, Y157	Near C-terminus of α1
Cluster 8	G13, Y32, N86, K117, GppNHp	Between P-loop and switch I

**Table 4 molecules-27-05710-t004:** CADD applications in RAS-related structure and binding sites identification.

CADD Methods	Results	References
Homology modeling	The 3D structure of RASSF2	[51]
Molecular dynamics simulation	The stability of the prediction model	[53]
Template-based protein–protein complex structure prediction algorithm (PRISM)	The structure of KRAS4B-GTP homodimer	[55]
AlphaFold	Models of 145 RAS superfamily members	[57]
Web server (Sitehound-Web)	Top 10 binding pockets on RASSF2	[51]
Probe-based molecular dynamics (PMD) simulation	Five potential druggable sites (S1–S3, Subsite 1 and Subsite 2) on KRAS	[53]
Fragment-based approach (FTMAP)	Three potential allosteric sites (P1–P3) on RAS	[67]
Multiple solvent crystal structures (MSCS)	Eight potential binding sites (Cluster 1–Cluster 8) on HRAS	[69]

**Table 5 molecules-27-05710-t005:** CADD applications in RAS inhibitor discovery.

Targeting Strategy	Drug	Targeting Information	CADD Methods			Reference
			Virtual Screening	Ligand-Based	Receptor-Based	
Direct targeting KRAS	Andrographolide (AGP) and its benzylidene derivatives	Binding to a transient pocket on KRAS, blocking GDP–GTP exchange			Molecular docking; Molecular dynamics	[91]
	Auriculasin	Blocking iKRAS^G12D^–SOS1 interaction, inhibiting the guanylate cycle		Similarity searching; Pharmacophore modelling (via ligand–receptor complex fingerprint)	Molecular docking; Molecular dynamics	[76]
	ARS-853, ARS-1620	Targeting the SII-P of RAS proteins in the GDP-bound state formation, interfering with the region of Switch 1 and Switch 2, blocking SOS-mediated GTP binding and effector proteins involvement,	**√**		Molecular docking	[44]
	Compound **D14** and **C22**	stabilizing the KRAS4B–PDE6δ molecular complex, and blocking the release of abnormal KRAS with mutations	**√**		Molecular docking; Molecular dynamics	[82]
Indirect targeting KRAS	IMB-1406	Inducing apoptosis in HepG2 cells by arresting the cell cycle at the S phase and altering anti- and pro-apoptotic proteins leading to mitochondrial dysfunction and activation of caspase-3, one of the possible targets being protein farnesyltransferase			Molecular docking	[132]
	NHTD	disrupting KRAS–PDEδ interaction, redistributing the localization of KRAS to endomembranes by targeting the prenyl-binding pocket of PDEδ	**√**			[80]
	Antroquinonol	Inhibiting prenyltransferase activity, blocking RAS and RAS-related GTP-binding protein activation	**√**		Molecular docking	[93]
	Theaflavin	Targeting farnesyltransferase, inhibiting PTM process			Molecular docking; Molecular dynamics	[78]
Upstream signaling pathway	Daidzein	Interacting with the kinase domain of the EGFR protein	**√**		Molecular docking; Molecular dynamics	[133]
	Scopoletin	Iargeting EGFR, BRAF, and AKT1 in NSCLC			Molecular docking	[134]
Downstream signaling pathway	Purine-2,6-dione analogues	Inhibiting BRAF protein kinase (a molecule in the RAS–RAF–MEK–ERK signaling pathway)			Molecular docking	[94]
	phosphoaminophosphonic acid adenylate ester (ANP), phosphoaminophosphonic acid guanylate ester (GNP)	Stabilizing RASSF2 (a KRAS-specific effector protein, promoting apoptosis and cell cycle arrest)	**√**		Molecular docking	[51]
	newly designed 2,6-disubstituted pyrazine derivatives	Inhibiting V600E BRAF		QSAR	Molecular docking (for the consideration of the similarity and alignment)	[125]
	Dehydrocoelenterazine	Interacting with the RAF kinase inhibitor protein (RKIP) ligand-binding pocket, thus inhibiting RKIP	**√**	Pharmacophore Modelling	Molecular docking; Molecular dynamics	[130]
	NCI 94680NCI 527880NCI 183519	BRAF inhibitor	**√**	QSAR; Pharmacophore modeling (used in the structural alignment step of QSAR modelling)	Molecular docking	[131]
	Pictilisib	PI3K-α inhibitor	**√**	Pharmacophore Modelling	Molecular docking	[59]
	Staurosporine	PKC-η inhibitor	**√**	Pharmacophore Modelling	Molecular docking	[59]
	Compound **M4**	MEK1 inhibitor	**√**	Pharmacophore Modelling	Molecular docking	[83]
	Catechin	MEK1 inhibitor	**√**	Similarity searching	Molecular docking (using the drug library obtained from similarity searching); Molecular dynamics	[84]
	CID-20759629	PI3Kγ/AKT/mTOR pan-inhibitor	**√**	Similarity searching	Molecular docking; Molecular dynamics	[135]
	Compound **17**	mTOR inhibitor	**√**	Similarity searching	Molecular docking; Molecular dynamics	[136]

## Data Availability

Not applicable.

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
