# Peer review of "Computer-Aided Drug Design Boosts RAS Inhibitor Discovery"

_molecules, 2022, doi:10.3390/molecules27175710_

Round 1

Reviewer 1 Report

In this review article, the authors provide an overview of the various computer-aided drug design (CADD) approaches utilized to identify RAS inhibitors. This is a well-written article with an in-depth explanation of the different CADD techniques as well. The reviewer recommends it for publication in the journal after addressing the following suggestions:

1. The title can be changed to a more appealing one. The word ‘help’ reads very colloquially.

2. Figure 1: label the HVR, G domain.

3. The authors have not discussed the role of the P-loop that is labeled in Figure 1.

4. A figure should be added to complement the mutations discussed in section 2.2. RAS mutations trigger signaling dysfunction.

5. In section 3.1, the authors mention ‘there are 7530-structures of RAS in the PDB.’ They then go on to describe the homology modeling studies of the cancer suppressor protein, RAS-association domain family (RASSF) 2. A question arises if there are crystal structures present in the RAS protein, why was this homology modeling study done. The authors should provide an explanation of why this study used homology modeling.

6. The last 2 paragraphs in section 3.1 have nothing to do with ras structure determination and should be either moved to the general introduction or removed.

7. Figure indicating the different sites that were identified should be included in section 3.2.

Reviewer 2 Report

The authors have conducted a review on the contribution of the computer-aided drug design (CADD) methods in the field of RAS inhibitors discovering. They first presented the members of the RAS family and their related signalling pathways, followed by an exposure of the dysfunctions generated by the RAS mutations.  The next part is dedicated to a well-detailed presentation of the             CADD methods and their applications for the identification of RAS inhibitors. They conclude by highlighting the great benefit of such methodologies in completion of classical experiments to study « undruggable targets » as RAS.

The manuscript is well-written and well-documented and can bring interesting information for RAS drug design. However some point need to be specified or detailed.

The Introduction paragraph is very short and what surprising me is that the references are relatively old (before 2014) apart the first two ones. Notably for the two main tactics mentioned, the vHTS reference date from 2012 and the de novo one from 2010. It appears to me that more recent publications may be profitably cited here, as for vHTS: doi:10.3389/fchem.2020.00343,10.2174/1568026619666190816101948, 10.1021/acs.jcim.0c01009 

and for de novo design: doi: 10.3390/ijms22041676, 10.1016/j.drudis.2021.05.019, 10.1002/anie.201814681

Considering the second strategy, de novo design method, I found surprising that it is cited in the introduction as a main tactic but only briefly mentioned at the end of the manuscript (l. 568 to 572), thus it seems not really adapted to the description of in silico RAS drug design.

For the first tactic, the authors claim (l. 51 to 53) "Structure-based vHTS requires molecular docking to critically evaluate ligand-receptor affinity and simulate their binding patterns while screening specific biologically active compounds. However, ligand-based vHTS is better suited to screen biologically active compounds ». Can they developed this concept? I have some difficulty to understand why LB vHTS would be better suited to screen biologically active compounds than SB vHTS. What the authors exactly mean by « ligand-based vHTS » in fact? I think this part need to be rewritten to be more explicit.

The second paragraph aims to describe the biological context of RAS inhibitor design. It seems that the figures may be improved to support the description of the RAS mechanism. The authors gave a structural description of RAS, including secondary structures as beta strands and alpha-helices and motifs as switch I/II and HVR region but this can not be easily seen on Figure 1. I think that a figure with only 3D structure of RAS is missing. 

I have also some difficulties to connect the Figure 1 and the Figure 2. On Figure 1, there is two  way between Res-GDP and Ras-GTP, one including GAP and the other SOS. On figure 2, only GAP appears… where is SOS? On Figure 1 there is an arrow conducting to Ras-RafRBD but I could not see this part of the scheme on Figure 2. On Figure 2, I could not understand what determine the MAPK or the PI3K-Akt signaling pathway. Moreover, the text in black on dark blue squares is not easy to read. Therefore it seems that more explanation and clearer figures are required for non biologist readers. For example, in part 2.2, l. 104, the authors mentioned « the arginine finger of GAP » but they never explained before what is this motif and its role.

The third, and the longest, paragraph deals with the CADD usage for the drug design of RAS inhibitors. This paragraph is divided into  8 parts following the several steps required to conduct such studies.

Part 3.1:

l. 143, 144: the authors assert that 7530 RAS structures are available in the RCSC PDB. That sounds a lot for me and I would like to know how they obtained this number. As a test, I have searched on the PDB web page the  term « RAS » first using the Full text search and I obtained 7425 results. Among these results, it appears lots of structure of proteins as p53 (5AOM, …), DJ-1 (6AFD, …) , Rheb (7BTC, …), and so on that can not be consider, in my opinion, as RAS structures. The same search by specifying the term « RAS » in the « Macromolecule name » field reduce the number of results to 1094 and to only 488 structures is the filter field is « Polymer Entity Description ». Thus I suggest to authors to check this number of available structures and to give details about the process used to evaluate this number.

At the end of this part, the authors mentioned the field of AI as methodology usable in future for protein structure prediction but, since the deployment of AlphaFold in 2021, it appears that is already the case. Perhaps the authors may develop the gain provided by such methodology to obtain the RAS structures.

Part 3.2:

The authors show a study on RASSF2 with specifying that «  the potential binding sites weere not known at that time » (l. 204). What about today? Is there more information? A comparison is it possible?

l. 244: the author said that AllositePro « incorporates protein dynamics into the training model for prediction », may they said in what way?

Part 3.3:

As in the introduction, the references used to introduce VS methods are too old in my opinion.

Part 3.4:

I find the reference 94 (l. 344) unadapted to introduce the general technic of molecular docking, since it refers to a specific software of docking. Some reviews as doi: 10.2174/157340911795677602 would be more appropriate. L. 356, the reference [97] used to justify the induced-fitting theory is very old (1968) and have to be updated.

Part 3.5:

I do not understand the sentence l. 447 to 449: "As second-generation force fields, MMF94 and CFF95 can be used to construct more complex functions and improve simulation accuracy, but this requires more computing power». What does the authors mean by « construct more complex functions »? For all I know, these force-field are used for the parameterisation of small molecules and it also exists in Amber and CHARMM such force-field dedicated to organic compounds in order to perform protein-ligand complexes simulations (GAFF in Amber and CGenFF in CHARMM). I can not see the reason justifying this part of the sentence "but this requires more computing power ». Moreover, the traditional force-field of CHARMM and Amber consider approximately the same entities (nucleic acids and amino acids) and may be used for approximately the same purpose.  I am not sure pretending that "The traditional supporting force fields AMBER are mainly used to simulate MD for polysaccharides, nucleic acids and small proteins. CHARMM is used to incorporate quantum calculations to expand the application scenarios, e.g., for apo structures and polymers in solution. » is really correct.

Part 3.6:

I find that the text l. 553 to 539 has to be placed before talking about the QSAR model for RAS study that started l. 523. The sentence "Overall, QSAR research needs structural alignment of compounds as the most critical step. » is not true, this is only the case for 3D-QSAR. This point is already discussed for CoMFA l. 519-520 so the authors have to merge the two paragraphs.

Part. 3.7:

l. 541-542: the term « pharmacophore » is well-known in the chemoinformatic field and I guess that it is possible to find a reference explaining the concept more adapted than reference [120]

Minor points:

- Standardise the name of Modeller software in the manuscript (see l. 151, 157, 396, …) 

- l. 175: remove the « . » after discovered

- l. 229: replace « identify » by « to identify »

- l. 296: replace « to binding to » by « to bind to »

- l. 431: replace « The » by « the » in « The kinematic »

- l. 476: replace « dimmer » by « dimer »

- l. 480: remove the first occurence of « simulation » in « In a 200-ns simulation MD simulation »

Round 2

Reviewer 1 Report

The authors have incorporated all the major edits. The manuscript is well written and can be accepted for publication as is.